# The Significance of Systemic Inflammation Markers in Intrahepatic Recurrence of Early-Stage Hepatocellular Carcinoma after Curative Treatment

**DOI:** 10.3390/cancers14092081

**Published:** 2022-04-21

**Authors:** Bong Kyung Bae, Hee Chul Park, Gyu Sang Yoo, Moon Seok Choi, Joo Hyun Oh, Jeong Il Yu

**Affiliations:** 1Department of Radiation Oncology, Samsung Medical Center, Sungkyunkwan University School of Medicine, Seoul 06351, Korea; bongkyung.bae@samsung.com (B.K.B.); gs.levin.yoo@samsung.com (G.S.Y.); 2Department of Medicine, Samsung Medical Center, Sungkyunkwan University School of Medicine, Seoul 06351, Korea; drms.choi@samsung.com (M.S.C.); 20210372@eulji.ac.kr (J.H.O.)

**Keywords:** hepatocellular carcinoma, intrahepatic recurrence, systemic inflammatory marker, neutrophil-to-lymphocyte ratio, platelet-to-lymphocyte ratio, lymphocyte-to-monocyte ratio

## Abstract

**Simple Summary:**

This retrospective study using the prospectively collected registry data of newly diagnosed, previously untreated hepatocellular carcinoma (HCC) evaluates the significance of systemic inflammatory markers (SIMs) to intrahepatic recurrence (IHR) after curative treatment. Out of 4076 patients who met the inclusion criteria, 52.6% experienced IHR. SIMs, including pre-treatment platelet-to-lymphocyte ratio (PLR), post-treatment changes of neutrophil-to-lymphocyte ratio PLR, and lymphocyte-to-monocyte ratio were significantly associated with the prognosis of early-stage HCC patients who received initial curative treatment. The prognostic significances of SIMs were consistent for IHR-free survival, early and late IHR, and overall survival.

**Abstract:**

Systemic inflammatory markers (SIMs) are known to be associated with carcinogenesis and prognosis of hepatocellular carcinoma (HCC). We evaluated the significance of SIMs in intrahepatic recurrence (IHR) of early-stage HCC after curative treatment. This study was performed using prospectively collected registry data of newly diagnosed, previously untreated HCC between 2005 and 2017 at a single institution. Inclusion criteria were patients with Barcelona Clinic Liver Cancer stage 0 or A, who underwent curative treatment. Pre-treatment and post-treatment values of platelet, neutrophil, lymphocyte, monocyte, neutrophil/lymphocyte ratio (NLR), platelet/lymphocyte ratio (PLR), and lymphocyte/monocyte ratio (LMR) were analyzed with previously well-known risk factors of HCC to identify factors associated with IHR-free survival (IHRFS), early IHR, and late IHR. Of 4076 patients, 2142 patients (52.6%) experienced IHR, with early IHR in 1018 patients (25.0%) and late IHR in 1124 patients (27.6%). Pre-treatment platelet count and PLR and post-treatment worsening of NLR, PLR, and LMR were independently associated with IHRFS. Pre-treatment platelet count and post-treatment worsening of NLR, PLR, and LMR were significantly related to both early and late IHR. Pre-treatment values and post-treatment changes in SIMs were significant factors of IHR in early-stage HCC, independent of previously well-known risk factors of HCC.

## 1. Introduction

Primary liver cancer was the sixth most commonly diagnosed cancer and the third leading cause of cancer death worldwide in 2020, and hepatocellular carcinoma (HCC) accounts for 75% to 85% of all primary liver cancers [1]. HCC occurrence is closely related to underlying chronic liver disease, which is usually caused by the hepatitis B virus, the hepatitis C virus, alcohol abuse, and metabolic-associated fatty liver disease (MAFLD) [1,2,3]. Due to the coexistence of underlying liver disease and HCC, potentially curative treatment options are available for selected patients with early HCC stages [4]. However, recurrence after treatment is common even with potentially curative treatment. Except for liver transplantation, more than half of patients develop recurrence during follow-up with an annual incidence of ≥10% [5,6,7,8].

Intrahepatic recurrence (IHR) of HCC is believed to occur because of two main factors. One is the aggressiveness and/or extent of the primary tumor itself, and the other is the carcinogenic environment of background liver due to underlying chronic liver disease [9,10]. Generally, the aggressiveness and/or extent of the primary tumor is associated with early IHR, and the carcinogenic environment is associated with late IHR [10]. Understanding and finding factors of respective recurrence patterns could help predict patient prognosis and improve outcomes by modifying these factors.

Liver inflammation is known to be associated with MAFLD, non-alcoholic steatohepatitis, fibrosis, and cirrhosis, leading to liver carcinogenesis [11]. Various systemic inflammatory markers (SIMs) have been widely studied to determine their correlation with the prognosis of HCC patients, and their prognostic value has been revealed [12,13,14,15,16]. However, the potential relationship between SIMs and IHR patterns in early-stage HCC patients after curative treatment has not been studied and is still unknown.

Therefore, we investigated the clinical significance of SIMs for IHR in early-stage HCC patients treated with curative treatment.

## 2. Materials and Methods

### 2.1. Patients

This was a retrospective study using prospectively collected registry data of newly diagnosed, previously untreated HCC from between January 2005 and December 2017 at the Samsung Medical Center. The inclusion criteria of the current study were as follows: (1) histologically or clinically confirmed HCC based on the guidelines of the Korean Liver Cancer Association-National Cancer Center [17,18,19], (2) early-stage HCC, the Barcelona Clinic Liver Cancer (BCLC) stage 0 or A, and (3) initial curative treatment. Patients who underwent liver transplantation, hepatic resection, and ablative treatments such as radiofrequency ablation (RFA), transarterial chemoembolization (TACE) plus RFA, TACE plus radiotherapy, and curative-aim radiotherapy were considered to have received initial curative treatment and were included in the current study.

### 2.2. Data Collection

We retrospectively collected the baseline characteristics of patients, tumors, and laboratory data. Collected laboratory data were obtained from the complete blood cell count and differential counts. Pre-treatment and post-treatment values of platelets, neutrophils, lymphocytes, monocytes, neutrophil-to-lymphocyte ratio (NLR), platelet-to-lymphocyte ratio (PLR), lymphocyte-to-monocyte ratio (LMR), and aspartate aminotransferase (AST)-to-lymphocyte ratio index (ALRI) were collected. NLR was calculated by dividing neutrophil count by lymphocyte count, PLR was calculated by dividing platelet count by lymphocyte count, LMR was calculated by dividing lymphocyte count by monocyte count, and ALRI was calculated by dividing AST value by lymphocyte count. Changes in values were determined by dividing the 3-to-6-month post-treatment values by the pre-treatment values.

### 2.3. Follow-Up

After treatment, all patients underwent computed tomography (CT) or magnetic resonance imaging (MRI) of abdomen and liver; and biochemical tests including complete blood cell count, liver function tests, and serum alpha-fetoprotein (AFP). If first follow-up imaging showed no evidence of residual tumor, follow-up examinations were done every 3 months in the first year, and every 3 to 6 months thereafter. Follow-up CT and MRI were thoroughly reviewed to evaluate IHR after initial curative treatment. The first IHR on the imaging study was counted. IHR-free survival (IHRFS) was defined as the time interval from initial curative treatment to the first IHR or death without IHR, whichever came first. Early and late IHRs were categorized on the basis of the time interval between the initial curative treatment and IHR. However, there is no clear definition of early recurrence, and early recurrence ranges from 6 months to 2 years between studies [9,20,21,22,23]. Yamamoto et al. calculated the optimal cut-off value with the “minimum *p*-value” approach and suggested that the most significant cut-off value was 17 months [24]. Based on the result of that study, we defined early recurrence as an IHR within 1.5 years from curative treatment, and late recurrence as an IHR after 1.5 years from curative treatment. Overall survival (OS) was defined as the time interval from the initial curative treatment to the date of death or the latest follow-up visit.

### 2.4. Study Endpoints

The primary and secondary endpoints of the current study were IHRFS and OS. The primary purpose was to evaluate the significance of SIMs in IHR after curative treatment, even when evaluated with previously well-known prognostic factors of HCC as covariates. The secondary purpose was to evaluate whether the SIMs that were significant in IHR also affected OS.

### 2.5. Statistical Analysis

Chi-square tests were performed to compare categorical variables. Before comparing continuous variables, we examined the normality of variables with a Kolmogorov–Smirnov test. A *t*-test was performed for factors following normal distribution, and a Wilcoxon rank sum test was performed for factors that did not follow normal distribution. SIMs were evaluated to find a cutoff point that best separated patients into two different groups with the most different IHRFS. Optimal cutoff values of SIMs were determined using maximally selected chi-square statistics [25]. This is a statistical method that determines the strength of grouping, selecting the cutoff point that minimizes the *p* value of the log-rank test. The results of the maximally selected chi-square statistics are shown in Appendix A. Cox regression analysis was performed for univariable and multivariable analyses of potential prognostic factors of IHRFS. Binary logistic regression analysis was performed to determine the factors associated with early and late IHR. Variables with a variance inflation factor < 4.0 were considered as candidate factors in multivariable models. The final multivariable model was determined using a backward variable selection method with elimination criteria of 0.05. OS was calculated using the Kaplan-Meier method and comparisons between groups were performed using the log-rank test. A *p*-value of <0.05 was considered to be statistically significant. Statistical analyses were performed using IBM SPSS Statistics ver. 22.0 (IBM, Inc., Armonk, NY, USA), R ver. 4.0.3, and R studio ver. 1.3.1093 (R Foundation for Statistical Computing, Vienna, Austria; http://www.r-project.org, accessed on 10 October 2020).

## 3. Results

Between January 2005 and December 2017, 9132 patients entered the HCC registry. Of the 9132 patients in the registry, 5506 patients had BCLC stage 0 or A. From these early-stage HCC patients, 1430 patients were excluded from the current study because patients did not receive initial curative treatment (1134 patients) or follow-up imaging studies were not available (296 patients). In total, 4076 patients with BCLC 0 or A who received initial curative treatment were included in the current study (Figure 1). The baseline clinical characteristics of patients with and without IHR are shown in Table 1.

IHR was observed in 2142 patients (52.6%). Median IHRFS was 4.9 years with a 2-year IHRFS of 68.3%, and 5-year IHRFS of 49.5%. The median time to IHR was 1.6 years (range 0.1–13.4 years), with peak incidence observed 0.6 years after initial curative treatment (92 recurrences). Early IHR, recurrence within 1.5 years from the initial curative treatment, occurred in 1018 patients, and late IHR, recurrence 1.5 years after the initial curative treatment, occurred in 1124 patients. Figure 2 shows the number of IHRs observed over time and the Kaplan-Meier survival curve.

Table 2 shows the results of the univariable, and multivariable analyses of potential prognostic factors associated with IHRFS. In the multivariable analysis, male sex, old age, high albumin-bilirubin (ALBI) grade, multiple tumors, and large tumor size were independent prognostic patient or tumor factors for IHRFS. Along with the previously well-known factors associated with HCC prognosis, several SIMs, including low pre-treatment AMC (*p* < 0.001, hazard ratio (HR) 1.206, confidence interval (CI) 1.088–1.337), low pre-treatment PLR (*p* < 0.001, HR 1.297, CI 1.187–1.417), high pre-treatment ALRI (*p* < 0.001, HR 1.550, CI 1.384–1.735), post-treatment increase in NLR (*p* < 0.001, HR 1.416, CI 1.280–1.566), post-treatment increase in PLR (*p* < 0.001, HR 1.889, CI 1.706–2.092), and post-treatment decrease in LMR (*p* < 0.001, HR 1.687, CI 1.530–1.860) were independently associated with IHRFS.

Table 3 shows the results of the binary logistic regression analysis to identify factors related to early and late IHR. Male sex and high ALBI grade were independent factors related to both early and late IHR. Multiple tumors and large tumor size were independent factors related to early IHR. High Child-Pugh score was related to late IHR. SIMs, including low ANC (*p* = 0.006, odds ratio (OR) 1.339 for late IHR), high pre-treatment NLR (*p* = 0.003, OR 1.430 for late IHR), low pre-treatment PLR (*p* < 0.001, OR 1.428 for early IHR; *p* < 0.001, OR 1.607 for late IHR), low pre-treatment ALRI (*p* < 0.001, OR 2.304 for early IHR; *p* < 0.001, OR 1.990 for late IHR), post-treatment increase in NLR (*p* < 0.001, OR 2.165 for early IHR; *p* < 0.001, OR 1.977 for late IHR), post-treatment increase in PLR (*p* < 0.001, OR 2.940 for early IHR; *p* < 0.001, OR 2.832 for late IHR), and post-treatment decrease in LMR (*p* < 0.001, OR 2.543 for early IHR; *p* < 0.001, OR 2.345 for late IHR), were significantly associated with IHR, even when previously well-known patient and tumor factors were analyzed together as covariates.

Figure 3 shows the Kaplan–Meier survival curves of IHRFS and OS with patients categorized based on SIMs which showed prognostic ability in IHR. All SIMs separated the survival curves considerably, except pre-treatment PLR which showed marginal significance for OS. The values of IHRFS and OS are shown in Table 4.

IHRFS, early IHR, and late IHR were also analyzed on the basis of patients who received local curative treatments recommended in multiple guidelines: hepatic resection and RFA [4,17,26,27,28]. These treatment options were delivered to 3858 patients; 2261 patients were treated with hepatic resection, and 1597 patients were treated with RFA. Similar findings were observed in the subgroup analysis. Low pre-treatment platelet count, high pre-treatment ALC, low pre-treatment AMC, low pre-treatment PLR, post-treatment increase in NLR, post-treatment increase in PLR, and post-treatment decrease in LMR were independently associated with IHRFS and early and late IHR (Appendix A). We further analyzed according to received initial treatment: hepatic resection and RFA. The results of Cox regression and binary logistic regression analyses of IHRFS, early IHR, and late IHR also showed the significance of SIMs for each method. The results for patients who underwent hepatic resection are summarized in Appendix A; and the results of patients who underwent RFA are summarized in Appendix A, respectively.

## 4. Discussion

Our study has shown that SIMs, including pre-treatment PLR and post-treatment changes in NLR, PLR, and LMR, had significant prognostic ability for IHRFS in early-stage HCC patients who received initial curative treatment. These factors were also significantly associated with both early and late IHRs. The prognostic values were independent of the previously well-known prognostic factors of HCC.

Inflammation is generally accepted to play a critical role in cancer development and progression [29,30,31]. From this point of view, SIMs are widely investigated for their clinical significance in various cancers, including HCC. Numerous studies have confirmed the importance of NLR, PLR, and LMR, reporting that high NLR, PLR, and low LMR are associated with worse outcomes [13,15,16,32]. Our results are in line with the other previously published studies, as SIMs were found to be important in the prognosis of HCC; however, some differences are present.

Whereas other studies reported that pre-treatment values of NLR, PLR, and LMR are important in prognosis, our results showed that only pre-treatment PLR was a significant factor of IHR. In addition, unlike other studies, our results showed that a low pre-treatment PLR value (<65) was associated with poor prognosis. This difference may be due to differences in the study population. Most studies on the value of SIMs have been conducted on patients with all stages of HCC, with various levels of underlying liver function. However, the population of the current study was patients with early-stage HCC who received initial curative treatment; therefore, they had relatively good baseline liver function, as shown in Table 1. When PLR 150 was selected as the cut-off point, the results were in line with those of other studies: no difference in IHFFS and worse OS in the high PLR group (*p* = 0.599 and *p* = 0.004, respectively; Appendix A). This could explain the relatively lower importance of pre-treatment SIMs in the current study compared with other published studies.

Rather than pre-treatment values, post-treatment changes in NLR, PLR, and LMR were associated with an increased risk of IHR. Although the values of pre-treatment SIMs have been widely studied and their importance has been elucidated, values of post-treatment changes of SIMs have been relatively less studied, and only a few articles have been published on various cancers. Zhuang et al. reported that high post-treatment PLR and post-treatment increase in NLR were associated with poor prognosis in patients with small HCCs treated with SBRT [33]. Lin et al. reported that low post-treatment LMR was associated with poor prognosis in patients who underwent radical gastrectomy for gastric cancer, and the prognostic accuracy of the TNM staging system was significantly improved by incorporating post-treatment LMR [34]. Khunger et al. reported that high post-treatment NLR was associated with poor prognosis in patients who received nivolumab monotherapy for non-small cell lung cancer [35]. In the current study, post-treatment increase in NLR, post-treatment increase in PLR, and post-treatment decrease in LMR were significantly associated with poor prognosis, increased IHR, and worse OS. With other studies, the results of the current study show that changes in SIMs after treatment could be an important factor in predicting the prognosis of cancer patients.

We also analyzed the potential prognostic factors for early and late IHR. For patient and tumor characteristics, male sex, viral infection status, and poor ALBI grade were associated with increased early and late IHRs. High AFP level, multiple primary tumors, and large tumor size, which represents the aggressiveness of the primary tumor, were related to increased early IHR. Centonze et al. reported that after surgical resection of HCC, AFP, microvascular invasion, satellitosis, and R1 resection were associated with relapse-free survival [36]. In addition, there are numerous studies supporting the importance of these patient and tumor risk factors to IHR [10,20,23,37]. Our results support and further extend that such previously known risk factors play an important role in the IHR of early-stage HCC patients who received initial curative treatment.

For SIMs, post-treatment changes in NLR, PLR, and LMR were related to early and late IHR. The role of SIMs in early and late IHR has not been well studied, and its significance has not yet been established. Liu et al. reported that for patients who underwent curative hepatic resection for HCC, a high pre-treatment NLR was a significant factor for early recurrence, and a high pre-treatment AST-to-platelet ratio index was a significant factor for both early and late recurrence [38]. The purpose and design of our study and the study by Liu et al. were similar, but the results were quite different. SIMs are known to be significantly correlated with various factors, including the BCLC stage [39]. Whereas our study included only patients with BCLC 0 or A, 97 out of 223 patients (43.5%) had BCLC B or C in the study by Liu et al. As our study had a more uniform baseline patient population, it could be considered that there were fewer confounding factors that could influence SIMs. Therefore, the significance of SIMs themselves can better revealed and understood.

Although only liver transplantation, hepatic resection, and RFA are considered curative treatment options for HCC, we also enrolled patients who received TACE plus RFA, TACE plus radiotherapy, and curative-aim radiotherapy. In general, TACE plus ablative treatments show better outcomes than TACE alone, but with worse outcomes compared with curative treatments [40,41,42]. However, TACE plus ablative treatment was not inferior to hepatic resection for patients within the Milan criteria [43] and for patients with BCLC stage A or B after propensity score matching [44]. Multiple studies have reported that curative-aim radiotherapy, such as stereotactic body radiotherapy or proton beam therapy, showed comparable results to RFA in the meta-analyses [45] and prospective trials [46]. Based on these results, we considered TACE plus ablative treatments and curative-aim radiotherapy in the early-stage HCC as curative treatments. Our findings were consistent irrespective of whether the patient groups were included (Table 2 and Table 3) or excluded (Appendix A).

Predicting the risk of recurrence after curative treatment is a matter of primary interest. Ng et al. performed a retrospective cohort study of HCC patients who underwent curative hepatic resection, and developed a prediction score model for early IHR [47]. They concluded that with four simple risk factors (AFP, tumor size, multiple tumors or satellite nodules, and microvascular invasion), it was possible to predict high-risk groups for early IHR. The study also suggested that high-risk patients could be the target of adjuvant treatment and aggressive surveillance. SIMs, a simple inflammation-based index which can be easily calculated for HCC patients, could add additional valuable information in identifying the high-risk patients for IHR.

There were several limitations to the current study. Although it was conducted on a large number of patients from prospectively collected data, selection bias could have occurred because the registry was collected from a single-institutional tertiary referral hospital in Korea. In addition, SIMs could have been influenced by uncollected data, such as the presence of concurrent inflammatory status or medications that could have been used for patient management. Additionally, our study did not cover how the patients were managed after IHR. Details of further management would have added additional depth but were not available at the time of study. However, we believe that the large number of patients included and the relatively even study population of early-stage HCC strengthen the significance of the findings of the current study.

## 5. Conclusions

The results of the current retrospective study showed that SIMs, including pre-treatment PLR and post-treatment changes in NLR, PLR, and LMR, were significantly associated with the prognosis of early-stage HCC patients who received initial curative treatment. The prognostic significance of SIMs was consistent for IHRFS, early and late IHR, and OS. Prospective and multicenter studies are needed to confirm the present results.

## Figures and Tables

**Figure 1 cancers-14-02081-f001:**
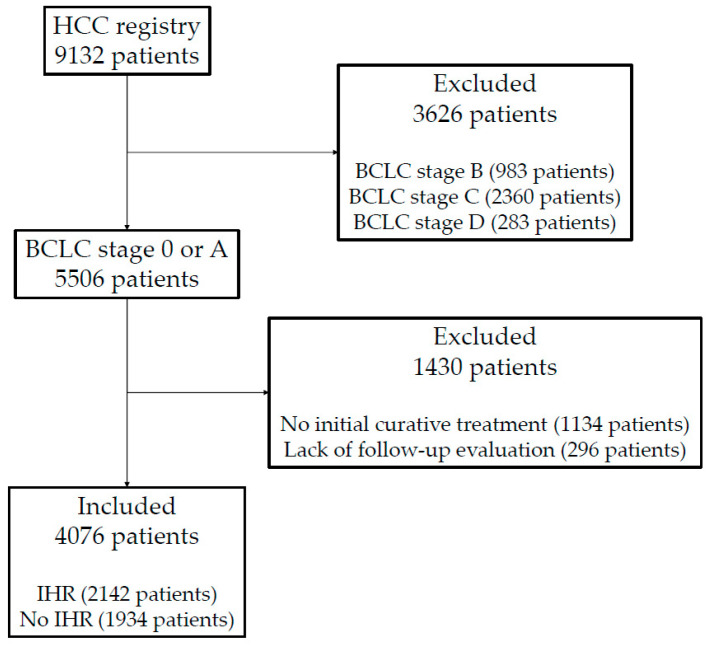
Flowchart of patient selection. HCC, hepatocellular carcinoma; BCLC, Barcelona Clinic Liver Cancer; IHR, intrahepatic recurrence.

**Figure 2 cancers-14-02081-f002:**
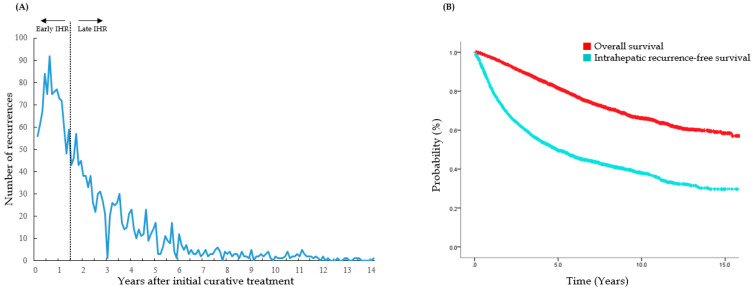
(**A**) Number of IHRs observed after initial curative treatment. The absolute counts of IHR are plotted in monthly bases. Recurrence within 1.5 years from initial curative treatment was grouped as early IHR, and recurrence after 1.5 years from initial curative treatment was grouped as late IHR. (**B**) Kaplan-Meier curves of OS and IHRFS. The 2-year and 5-year OS rates were 93.6% and 81.6%, respectively. The 2-year and 5-year IHRFS rates were 68.3% and 49.5%, respectively. IHR, intrahepatic recurrence; OS, overall survival; IHRFS, intrahepatic recurrence-free survival.

**Figure 3 cancers-14-02081-f003:**
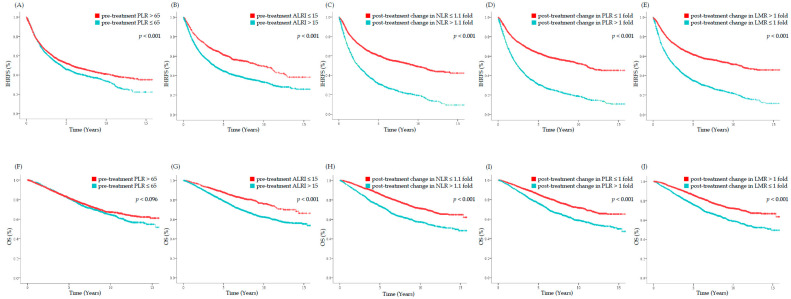
Kaplan–Meier curves of IHRFS and OS stratified by SIMs which showed prognostic significance to IHR: (**A**–**E**) IHRFS; (**F**–**J**) OS. IHRFS, intrahepatic recurrence-free survival; OS, overall survival; SIMs, systemic inflammatory markers; PLR, platelet-to-lymphocyte ratio; ALRI, aspartate-transaminase-to-lymphocyte ratio index; NLR, neutrophil-to-lymphocyte ratio; LMR, lymphocyte-to-monocyte ratio.

**Table 1 cancers-14-02081-t001:** Baseline clinical characteristics.

	IHR (+) *n* = 2142	IHR (−) *n* = 1934	*p*-Value
Sex			<0.001 *
Male	1720 (80.3%)	1465 (75.7%)	
Female	422 (19.7%)	469 (24.3%)	
Age			0.949 **
Mean (± SD)	57.2 (± 9.8)	57.3 (±10.4)	
Viral status			<0.001 *
Viral	1874 (87.5%)	1605 (83.0%)	
Non-viral	268 (12.5%)	329 (17.0%)	
ALBI grade			<0.001 *
1	1388 (64.8%)	1515 (78.3%)	
2	710 (33.1%)	382 (19.8%)	
3	44 (2.1%)	37 (1.9%)	
Child-Pugh Class			0.290 *
A	1975 (92.2%)	1800 (93.1%)	
B	167 (7.8%)	134 (6.9%)	
BCLC stage			0.551 *
0	777 (36.3%)	719 (37.2%)	
A	1365 (63.7%)	1215 (62.8%)	
AFP			<0.001 ***
Median (IQR)	16 (5.7–90.75)	11 (3.8–86.55)	
*n* of tumor			<0.001 *
1	1881 (87.8%)	1797 (92.9%)	
2	236 (11.0%)	125 (6.5%)	
3	25 (1.2%)	12 (0.6%)	
Tumor size			0.039 **
Mean (± SD)	2.6 (± 1.7)	2.7 (±1.8)	
T stage			<0.001 *
1	856 (39.9%)	783 (40.5%)	
2	1171 (54.7%)	1096 (56.7%)	
3	115 (5.4%)	55 (2.8%)	
Primary treatment			<0.001 *
Liver transplantation	4 (0.2%)	65 (3.4%)	
Resection	1020 (47.6%)	1241 (64.2%)	
Ablative treatment	1118 (52.2%)	628 (32.5%)	
RFA	1021 (47.7%)	576 (29.8%)	
TACE + RFA	79 (3.7%)	43 (2.2%)	
TACE + radiotherapy	13 (0.6%)	7 (0.4%)	
Curative radiotherapy	7 (0.2%)	2 (0.1%)	

IHR, intrahepatic recurrence; *n*, number; SD, standard deviation; IQR, interquartile range; ALBI, albumin–bilirubin; BCLC, Barcelona Clinic Liver Cancer; AFP, alpha-fetoprotein; RFA, radiofrequency ablation; TACE, transarterial chemoembolization. * Chi-square test; ** *t*-test; *** Wilcoxon rank sum test.

**Table 2 cancers-14-02081-t002:** Univariable and multivariable Cox regression analysis of intrahepatic recurrence-free survival.

Factors	Univariable		Multivariable	
	HR (95% CI)	*p*-Value	HR (95% CI)	*p*-Value
Sex (Male)	1.193 (1.073–1.327)	0.001	1.211 (1.084–1.353)	0.001
Age (>60)	1.108 (1.014–1.210)	0.024	1.241 (1.129–1.365)	<0.001
Viral status (Viral)	1.158 (1.018–1.316)	0.023		
AST (>40)	1.546 (1.420–1.684)	<0.001		
ALT (>40)	1.295 (1.187–1.413)	<0.001		
ALBI grade (2–3)	1.528 (1.398–1.670)	<0.001	1.157 (1.051–1.275)	0.003
Child-Pugh score (6–9)	1.302 (1.176–1.441)	<0.001		
AFP (>100)	1.036 (0.938–1.143)	0.488		
*n* of tumor (2–3)	1.562 (1.372–1.778)	<0.001	1.342 (1.177–1.532)	<0.001
Tumor size (>5 cm)	1.102 (0.957–1.268)	0.184	1.314 (1.112–1.552)	0.001
BCLC stage (A)	1.119 (1.025–1.222)	0.012		
Platelet (≤150,000)	1.493 (1.366–1.632)	<0.001		
ANC (≤2500)	1.265 (1.162–1.377)	<0.001		
ALC (>1200)	1.094 (0.985–1.215)	0.097		
AMC (≤500)	1.093 (0.993–1.203)	0.073	1.206 (1.088–1.337)	<0.001
NLR (>1.0)	1.019 (0.921–1.127)	0.718		
PLR (≤65.0)	1.174 (1.078–1.278)	<0.001	1.297 (1.187–1.417)	<0.001
LMR (≤4.0)	1.082 (0.971–1.206)	0.159		
ALRI (>15.0)	1.612 (1.458–1.782)	<0.001	1.550 (1.384–1.735)	<0.001
Change in NLR (>1.1 fold)	2.294 (2.107–2.498)	<0.001	1.416 (1.280–1.566)	<0.001
Change in PLR (>1 fold)	2.514 (2.307–2.740)	<0.001	1.889 (1.706–2.092)	<0.001
Change in LMR (<1 fold)	2.252 (2.066–2.455)	<0.001	1.687 (1.530–1.860)	<0.001
Change in ALRI (>1.1 fold)	1.217 (1.118–1.325)	<0.001		

AST, aspartate transaminase; ALT, alanine transferase; ALBI, albumin–bilirubin; AFP, alpha-fetoprotein; *n*, number; BCLC, Barcelona Clinic Liver Cancer; ANC, absolute neutrophil count; ALC, absolute lymphocyte count; AMC, absolute monocyte count; NLR, neutrophil-to-lymphocyte ratio; PLR, platelet-to-lymphocyte ratio; LMR, lymphocyte-to-monocyte ratio; ALRI, AST-to-lymphocyte ratio index.

**Table 3 cancers-14-02081-t003:** Univariable and multivariable results of binary logistic regression analysis for potential factors of early and late intrahepatic recurrence.

Factors	Early IHR				Late IHR			
	Univariable		Multivariable		Univariable		Multivariable	
	OR (95% CI)	*p*-Value	OR (95% CI)	*p*-Value	OR (95% CI)	*p*-Value	OR (95% CI)	*p*-Value
Sex (Male)	1.351 (1.120–1.630)	0.001	1.426 (1.162–1.749)	0.001	1.265 (1.058–1.513)	0.010	1.397 (1.122–1.738)	0.003
Age (>60)	0.936 (0.799–1.095)	0.408			0.931 (0.799–1.084)	0.355		
Viral status (Viral)	1.425 (1.144–1.776)	0.002			1.441 (1.164–1.783)	0.001		
AST (>40)	2.130 (1.821–2.490)	<0.001			1.673 (1.436–1.950)	<0.001		
ALT (>40)	1.684 (1.437–1.975)	<0.001			1.466 (1.254–1.713)	<0.001		
ALBI grade (2–3)	2.136 (1.807–2.524)	<0.001	1.456 (1.199–1.768)	<0.001	1.818 (1.542–2.142)	<0.001	1.605 (1.235–2.086)	<0.001
Child-Pugh score (6–9)	1.536 (1.273–1.853)	<0.001			1.374 (1.141–1.654)	0.001	1.365 (1.019–1.829)	0.037
AFP (>100)	1.224 (1.029–1.454)	0.022			1.171 (0.981–1.399)	0.081		
*n* of tumor (2–3)	2.196 (1.716–2.811)	<0.001	1.729 (1.314–2.274)	<0.001	1.495 (1.153–1.938)	0.002		
Tumor size (>5 cm)	1.089 (0.838–1.416)	0.524	1.385 (1.035–1.855)	0.029	0.867 (0.738–1.121)	0.648		
BCLC stage (A)	1.279 (1.089–1.503)	0.003			0.870 (0.748–1.012)	0.070		
Platelet (≤150,000)	1.797 (1.537–2.101)	<0.001			1.874 (1.610–2.181)	<0.001		
ANC (≤2500)	1.497 (1.284–1.745)	<0.001			1.386 (1.194–1.608)	<0.001	1.339 (1.088–1.647)	0.006
ALC (>1200)	0.918 (0.759–1.110)	0.378			0.911 (0.758–1.095)	0.323		
AMC (>500)	1.028 (0.863–1.223)	0.760			1.101 (0.931–1.301)	0.261		
NLR (> 1.0)	0.996 (0.828–1.200)	0.970			0.845 (0.709–1.006)	0.058	1.430 (1.130–1.809)	0.003
PLR (≤ 65.0)	1.232 (1.055–1.439)	0.008	1.428 (1.190–1.714)	<0.001	1.769 (1.494–2.095)	<0.001	1.607 (1.326–1.946)	<0.001
LMR (≤4.0)	1.013 (0.836–1.227)	0.897			1.096 (0.908–1.324)	0.338		
ALRI (>15.0)	2.100 (1.783–2.472)	<0.001	2.304 (1.880–2.825)	<0.001	1.697 (1.423–2.025)	<0.001	1.990 (1.599–2.476)	<0.001
Change in NLR (>1.1 fold)	3.808 (3.233–4.486)	<0.001	2.165 (1.786–2.623)	<0.001	3.081 (2.626–3.613)	<0.001	1.977 (1.588–2.461)	<0.001
Change in PLR (>1 fold)	4.370 (3.714–5.141)	<0.001	2.940 (2.443–3.537)	<0.001	3.583 (3.063–4.191)	<0.001	2.832 (2.315–3.466)	<0.001
Change in LMR (<1 fold)	3.588 (3.060–4.207)	<0.001	2.543 (2.120–3.051)	<0.001	2.735 (2.348–3.186)	<0.001	2.345 (1.921–2.863)	<0.001
Change in ALRI (>1.1 fold)	1.332 (1.154–1.537)	<0.001			1.118 (0.953–1.312)	0.171		

AST, aspartate transaminase; ALT, alanine transferase; ALBI, albumin–bilirubin; AFP, alpha-fetoprotein; *n*, number; BCLC, Barcelona Clinic Liver Cancer; PLT, platelet; ANC, absolute neutrophil count; ALC, absolute lymphocyte count; AMC, absolute monocyte count; NLR, neutrophil-to-lymphocyte ratio; PLR, platelet-to-lymphocyte ratio; LMR, lymphocyte-to-monocyte ratio; ALRI, AST-to-lymphocyte ratio index.

**Table 4 cancers-14-02081-t004:** IHRFS and OS of patients categorized by SIMs.

Factors	IHRFS			OS		
	2-Year	5-Year	*p*-Value	2-Year	5-Year	*p*-Value
Pre-treatment PLR			<0.001			0.096
>65.0	69.2%	52.0%		93.8%	81.1%	
≤65.0	67.0%	45.8%		93.3%	81.8%	
Pre-treatment ALRI			<0.001			<0.001
≤15.0	76.9%	61.2%		96.1%	87.8%	
>15.0	64.7%	44.5%		92.5%	79.0%	
Post-treatment change in NLR			<0.001			<0.001
≤1.1 fold	76.0%	60.0%		95.6%	86.1%	
>1.1 fold	54.7%	31.0%		89.8%	73.6%	
Post-treatment change in PLR			<0.001			<0.001
≤1 fold	77.4%	62.8%		95.1%	85.4%	
>1 fold	55.1%	30.7%		91.2%	76.1%	
Post-treatment change in LMR			<0.001			<0.001
>1 fold	77.1%	61.5%		95.4%	86.0%	
≤1 fold	57.4%	34.7%		91.2%	76.1%	

IHRFS, intrahepatic recurrence-free survival; OS, overall survival; SIMs, systemic inflammatory markers; PLR, platelet-to-lymphocyte ratio; ALRI, aspartate-transaminase-to-lymphocyte ratio index; NLR, neutrophil-to-lymphocyte ratio; LMR, lymphocyte-to-monocyte ratio.

## Data Availability

The data are not publicly available due to ethical issues.

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
