# Peer review of "The Significance of Systemic Inflammation Markers in Intrahepatic Recurrence of Early-Stage Hepatocellular Carcinoma after Curative Treatment"

_cancers, 2022, doi:10.3390/cancers14092081_

Round 1

Reviewer 1 Report

I read with much interest this original paper from Bae and coll. entitled “The significance of systemic inflammation markers in intrahepatic recurrence of early stage hepatocellular carcinoma after curative treatment”

Although well written and properly discussed, there are some poi ta that should be addressed before publication

1) TACE and Radiotherapy are not curative treatments: altough dis ussed, the Authors should exclude these patients from the analysis or at least provide the number of patients treated for each group - moreover, the higher prevalence of ablative treatments in IHR+ group could be justified by their lower efficacy

2) please provide a flowchart for patient selecion

3) better describe surveillance and recurrence treatment, as well as the prevalence of salvage transplantation

4) seceral continuos variables are accounted as dicotomic in multivariate model: how was these cutoffs derived?

5) Better discuss the impact of pathological features on tumor recurrence, referring to this recently published paper 10.3390/diagnostics12010160

Congeatulations to the Authors for their efforts

Best regards

Author Response

The manuscript has been rechecked, and appropriate changes have been made in accordance with the reviewers’ suggestions. The responses to their comments have been prepared and are given below.

We thank you and the reviewers for your thoughtful suggestions and insights, which have enriched the manuscript and helped us produce a better and more balanced account of the research. We hope that the revised manuscript is now suitable for publication in your journal.

Reviewer 1.

  1. TACE and Radiotherapy are not curative treatments: altough dis ussed, the Authors should exclude these patients from the analysis or at least provide the number of patients treated for each group - moreover, the higher prevalence of ablative treatments in IHR+ group could be justified by their lower efficacy

Thank you for your valuable suggestions. We will provide the number of patients treated for each group in table 1. And also, we further analyzed according to the generally accepted curative treatment: hepatic resection + RFA; hepatic resection; RFA. The results of analyses according to treatments are summarized and added in Supplementary Tables 3 to 6, and documented in Results section.

  1. please provide a flowchart for patient selection

We have added patient selection flowchart in Result section, Figure 1.

  1. better describe surveillance and recurrence treatment, as well as the prevalence of salvage transplantation

We have added the details of patient surveillance in the follow-up subsection in Materials and methods section. Management of recurrence was not reviewed for patients of current study. Data of salvage treatments and survival afterward would have added additional depth in our study, but was not available. We will address this point additionally in the limitations in the Discussion section.

  1. Several continuous variables are accounted as dicotomic in multivariate model: how was these cutoffs derived?

Cutoff values were selected with maximally selected chi square method. It is a statistical method determining the strength of grouping, which selects the cutoff point that minimizes p value of log rank test. Details of this method was not mentioned in the previous version of manuscript, ant we have added the details in the statistical analysis subsection in Materials and methods section.

  1. Better discuss the impact of pathological features on tumor recurrence, referring to this recently published paper 10.3390/diagnostics12010160

We have read the article you have suggested, and cited the article in current study.

Reviewer 2 Report

Intrahepatic failure play significantly part in pathogenesis in the oncological diseases such as hepatocellular carcinoma. Its prognosis can be important part of diagnostic methods Presented results proved, that intrahepatic failure and some studied markers are strongly associated with decrease in overall survival. Nevertheless, some point can be taken for the improvement of manuscript.  

Could authors show any graph/s correlation between tested markers and intrahepatic failure?

Figure 1A- If these cases are given in absolute values, the basis to which they relate should be stated.

Every abbreviation used in figure/table should be defined in its label.

Figure 2 – Authors should be showed Kaplan-Meier curves of OS

Below works should be cited and discussed in the manuscript.

Ng KK, Cheung TT, Pang HH, Wong TC, Dai JW, Ma KW, She WH, Kotewall CN, Lo CM. A simplified prediction model for early intrahepatic recurrence after hepatectomy for patients with unilobar hepatocellular carcinoma without macroscopic vascular invasion: An implication for adjuvant therapy and postoperative surveillance. Surg Oncol. 2019 Sep;30:6-12. doi: 10.1016/j.suronc.2019.05.017. Epub 2019 May 22. PMID: 31500787.

Author Response

  1. Could authors show any graph/s correlation between tested markers and intrahepatic failure?

Thank you for your valuable suggestions. We have added additional survival curves showing the intrahepatic recurrence-free survival and overall survival of patients in Figure 2. Patients were stratified based on the markers which showed prognostic significance to intrahepatic recurrence.

  1. Figure 1A- If these cases are given in absolute values, the basis to which they relate should be stated.

1A graph shows the absolute values of intrahepatic recurrences in monthly bases. The explanation was not in the manuscript, we have added the explanation in the legend of Figure 1.

  1. Every abbreviation used in figure/table should be defined in its label.

The definition of abbreviation used in figures were not included in the Figure legends. We re-checked the manuscript and added the missed information.

  1. Figure 2 – Authors should be showed Kaplan-Meier curves of OS

As in the answer to your first comment, we have added additional survival curves of intrahepatic recurrence-free survival and overall survival in Figure 2.

  1. Below works should be cited and discussed in the manuscript.

Ng KK, Cheung TT, Pang HH, Wong TC, Dai JW, Ma KW, She WH, Kotewall CN, Lo CM. A simplified prediction model for early intrahepatic recurrence after hepatectomy for patients with unilobar hepatocellular carcinoma without macroscopic vascular invasion: An implication for adjuvant therapy and postoperative surveillance. Surg Oncol. 2019 Sep;30:6-12. doi: 10.1016/j.suronc.2019.05.017. Epub 2019 May 22. PMID: 31500787.

We have read the work you have suggested, and added related contents in the discussion section.

Reviewer 3 Report

In the present study, Bae and collegues investigated the value of SIMs for the prediction of HCC recurrence in patients with a diagnosis of early HCC treated with curative approaches. Overall, the manuscript is interesting and well written. However, there are some issues that need to be addressed.

1) Authors may consider to include the assessment of ALRI score (AST to lymphocyte ratio index).

2) Materials and methods section. I suggets to split out the section in different subsections, such as: Patients, Study Endpoints, Ststistical Analysis.

3) Statistical analysis. Was data normality checked? Which statistical test was used?

4) Univariate and multivariate analysis. Continuous variables were dichotomized according to specific cut-offs; it is unclear to me how were these cut-offs selected/defined? Furthermore, I suggest to perform subanalysis according to treatment received. Otherwise, type of treatment should be included in regression analyses.

Author Response

  1. Authors may consider to include the assessment of ALRI score (AST to lymphocyte ratio index).

Thank you for your valuable suggestions. We have added pre-treatment and post-treatment changes in ALRI score and re-analyzed the statistics. Pre-treatment ALRI score showed statistical significance in intrahepatic recurrence free survival, early IHR, and late IHR. HR, OR, p-values of factors in multivariable analysis has changed marginally, but in general, the results were consistent.

  1. Materials and methods section. I suggets to split out the section in different subsections, such as: Patients, Study Endpoints, Ststistical Analysis.

We further subdivided the materials and methods section.

  1. Statistical analysis. Was data normality checked? Which statistical test was used?

Normality of continuous variables were checked with Kolmogorov-Smirnov test. T-test was used for comparison of factors following normal distribution, and Wilcoxon rank sum test was used for comparison of factors not following normal distribution. We have added explanation in the Statistical analysis subsection in Materials and methods section.

  1. Univariate and multivariate analysis. Continuous variables were dichotomized according to specific cut-offs; it is unclear to me how were these cut-offs selected/defined?

Cutoff values were selected with maximally selected chi square method. It is a statistical method determining the strength of grouping, which selects the cutoff point that minimizes p value of log rank test. Details of this method was not mentioned in the previous version of manuscript, ant we have added the details in the statistical analysis subsection in Materials and methods section.

  1. Furthermore, I suggest to perform subanalysis according to treatment received. Otherwise, type of treatment should be included in regression analyses.

We have performed additional subanalysis of patients according to treatment received: hepatic resection and RFA. The results were similar for all subgroups. SIMs were significantly related with intrahepatic recurrence free survival, early IHR, late IHR, independent from other well known risk factors. The results of subgroup analysis is summarized in Supplementary tables 3 to 6, and commented in the last paragraph of Results section.

Round 2

Reviewer 1 Report

The Authors properly revised their original paper according to reviewers comments

The current version is suitable for publication

Reviewer 3 Report

The manuscript has been improved according to the comments raised. In my opinion, the manuscript is now suitable for publication in Cancers